

# New and known free-living nematode species (Nematoda: Chromadorea) from offshore tsunami monitoring buoys in the Southwest Pacific Ocean

Daniel Leduc

Oceans Centre, NIWA, Wellington, WGN, New Zealand

## ABSTRACT

Deep-ocean Assessment and Reporting of Tsunami (DART) buoys are deployed across the Southwest Pacific and provide substrates for biofouling communities. Two new free-living nematode species, *Atrochromadora tereroa* sp. nov. and *Euchromadora rebeccae* sp. nov. (family Chromadoridae), and one known species, *Halomonhystera refringens* (Bresslau & Schuurmans Stekhoven, 1933) comb. nov. (family Monhysteridae), are described from buoys deployed off Raoul Island in the Kermadec/Rangitāhua region and off New Zealand's East Cape. *Thalassomonhystera refringens* (Bresslau & Schuurmans Stekhoven, 1933) Jacobs, 1987 and *T. anoxybiotica* (Jensen, 1986) Jacobs, 1987 are transferred to *Halomonhystera* based on the presence of precloacal and caudal papillae in males. In addition, *Halomohystera zhangi* Li, Huang & Huang, 2024 is synonymised with *Halomonhystera refringens*. Updated keys to *Atrochromadora*, *Euchromadora* and *Halomonhystera* species are provided. The presence of nematodes on buoys located more than 100 km from the nearest landmass and in deep waters (>3,500 m water depth) shows that some nematode species are capable long-distance dispersal to colonise new substrates. Such dispersal by *Atrochromadora*, *Euchromadora* and *Halomonhystera* species likely occurs via drifting macroalgal fragments.

## INTRODUCTION

Molecular studies have shown that, although genetic connectivity among nematode populations generally appears to be limited to distances of less than 100 km (*Derycke et al., 2008*; *Derycke, Backeljau & Moens, 2013*; *Hauquier et al., 2017*), there is nonetheless evidence of gene flow between nematode communities separated by hundreds of kilometres (*Bik et al., 2010*; *Apolonio Silva de Oliveira et al., 2017*; *De Groote et al., 2017*). Nematodes have increasingly been recognized as having high dispersal abilities despite their limited mobility and the absence of a pelagic larval stage, with dispersal thought to occur mainly through passive means including drifting, rafting, zoochory and human-mediated transport (*Cerca, Purschke & Struck, 2018*; *Ptatscheck & Traunsperger, 2020*). Artificial structures such as ship hulls provide a suitable substrate for a range of epibiotic nematodes, particularly once they are colonized by biofilm-forming microorganisms and/or habitat-forming macroalgae

Corresponding author
Daniel Leduc,
daniel.leduc@niwa.co.nz

and invertebrates (*Jensen, 1984*; *Kito & Nakamura, 2001*; *Fonseca-Genevois et al., 2006*; *Majdi et al., 2011*; *Leduc, 2020*). Settlement plate experiments have also demonstrated the ability of nematodes to colonise artificial hard substrates deployed several meters above the seafloor in coastal environments (*Fonseca-Genevois et al., 2006*; *Boeckner, Sharma & Proctor, 2009*; *von Ammon et al., 2018*).

Tsunami detection and early warning represents an international effort, with systems deployed across the world's oceans. Twelve locations in the Southwest Pacific Ocean have been identified as part of the New Zealand Tsunami Detection Network, with the first deployment voyage taking place in late 2019. Each of the deployed Deep-ocean Assessment and Reporting of Tsunami (DART) systems comprises two major components: (a) a bottom pressure recorder with associated bottom acoustic release/flotation, and (b) a surface buoy with associated mooring lines, acoustic release and weights. The DART systems are deployed for about 24 months before being serviced and replaced, and during this period the surface buoys can accumulate a significant amount of biofouling. The presence of these buoys in locations across the Southwest Pacific provides a unique opportunity to study the nematode fauna colonizing structures located in deep water and more than 100 km away from the nearest landmass. In this study, I describe two new species of the family Chromadoridae (*Atrochromadora tereroa* sp. nov. and *Euchromadora rebeccae* sp. nov.) and one known species of the family Monhysteridae (*Halomonhystera refringens* (Bresslau & Schuurmans Stekhoven, 1933) comb. nov.) from buoys deployed off Raoul Island in the Rangitāhua/Kermadec region and off New Zealand's East Cape.

## MATERIALS & METHODS

Biofouling community samples were obtained from DART Buoy C deployed approximately 150 km east of New Zealand's East Cape and from DART Buoy F deployed approximately 245 km east of Raoul Island in the Rangitāhua/Kermadec region (Figs. 1 & 2, Table 1). Rangitāhua is within the rohe (territory) of Ngāti Kuri, with the islands holding spiritual, cultural and customary significance (*Ngāti Kuri Trust Board, 2013*). As kaitiaki (guardians/stewards), Ngāti Kuri seek to understand and protect the biota dwelling on land and in the surrounding seas, regarding these organisms as taonga (treasures), and recognising the national and international significance of the unique diversity and assemblages found at Rangitāhua (*Leduc, 2024*). One of Ngāti Kuri's current priorities is the documentation of the species occurring within their rohe. The research reported here was undertaken in collaboration with Ngāti Kuri, who contributed to the scientific naming of Rangitāhua species through mātauranga Māori (Māori knowledge). Specimen collection was conducted under Ministry for Primary Industries Special Permit No. 666-9.

The entire biofouling community within a 0.1 × 0.1 m quadrat placed on the side of each buoy was carefully scraped off using a plastic paint scraper, transferred to a plastic jar and fixed in buffered 10% formalin. In the laboratory, samples were passed through a one mm mesh to remove large biota (*e.g.*, filamentous algae, gooseneck barnacles) and then through a 45 μm mesh to retain nematodes. Nematodes were then picked under a dissecting microscope, transferred to pure glycerol and mounted onto permanent slides (*Somerfield & Warwick, 1996*).

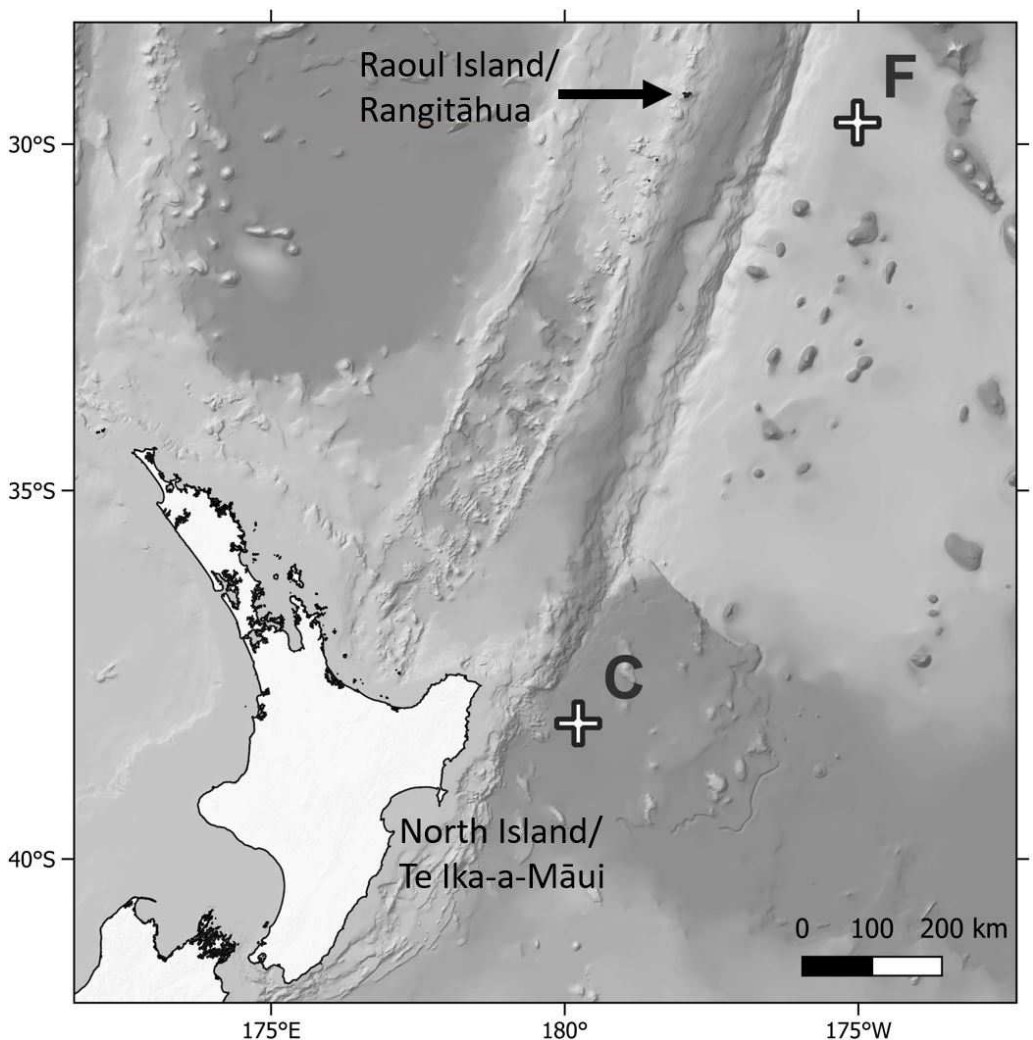

**Figure 1** **Map of sampling locations.** Map showing location of the Deep ocean Assessment and Reporting of Tsunami (DART) buoys C and F sampled in this study, in relation to New Zealand's North Island and Raoul Island.

Species descriptions were made from glycerol mounts using differential interference contrast (DIC) microscopy, and drawings were prepared with the aid of a camera lucida (*Leduc, 2023*). Measurements were obtained using an Olympus BX53 compound microscope with cellSens Standard software for digital image analysis. All measurements are in μm (unless stated otherwise), and all curved structures are measured along the arc. The terminology used to describe the arrangement of morphological features such as setae follows *Coomans (1979)*, and terminology for stoma structures follows *Decraemer, Coomans & Baldwin (2014)*. Type specimens are deposited in the NIWA Invertebrate Collection (Wellington).

The electronic version of this article in Portable Document Format (PDF) will constitute a published work according to the International Commission on Zoological Nomenclature
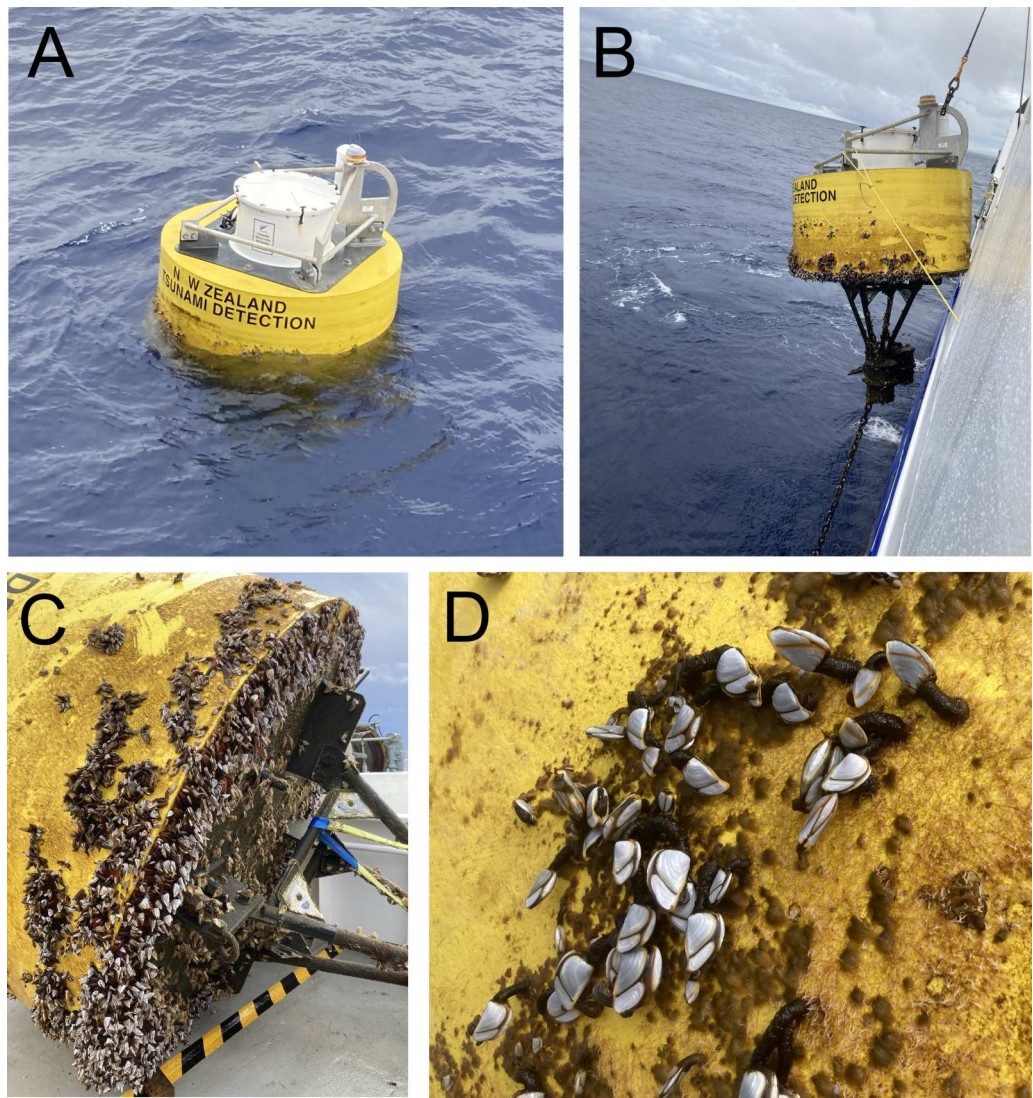

**Figure 2** **Deep ocean Assessment and Reporting of Tsunami (DART) buoys.** (A) Buoy F (Kermadec region) immediately prior to retrieval; (B) retrieval of buoy F; (C) buoy F immediately after retrieval, showing mix cover of filamentous algae and goose barnacles; (D) close up of buoy C (East Cape region) showing mixed cover of filamentous algae and goose barnacles.

**Table 1** **Details of the deep-ocean assessment and reporting of Tsunami (DART) buoys sampled in the present study.**

| Voyage | Station | Buoy | Latitude | Longitude | Water depth | Region | Buoy deployment date | Buoy collection date | Nematode species | Buoy epibiota |
|---|---|---|---|---|---|---|---|---|---|---|
| TAN2114 | DART 1 | C | −38.2002 | −179.7690 | 3,600 | East Cape | 12/2019 | 10/12/2021 | *Euchromadora rebeccae* sp. nov. *Halomohystera refringens* comb. nov. | Filamentous algae, goose barnacles |
| TAN2209 | DART 12 | F | −29.6782 | −175.0127 | 5,100 | Kermadec | 08/2021 | 10/08/2022 | *Atrochromadora tereroa* sp. nov. | Filamentous green algae |

(ICZN), and hence the new names contained in the electronic version are effectively published under that Code based on the electronic edition alone. This published work and the nomenclatural acts it contains have been registered in ZooBank, the online registration system for the ICZN. The ZooBank LSIDs (Life Science Identifiers) can be resolved, and the associated information viewed through any standard web browser by appending the LSID to the prefix http://zoobank.org/. The LSID for this publication is: urn:lsid:zoobank.org:pub:12C307BD-8D44-492C-AB65-673315A31097. The online version of this work is archived and available from the following digital repositories: PeerJ, PubMed Central, and CLOCKSS.

# RESULTS

Phylum Nematoda Cobb, 1932
Class Chromadorea Inglis, 1932
Order Chromadorida Chitwood, 1933
Family Chromadoridae Filipjev, 1917

**Family diagnosis (from *Tchesunov (2014)*)** Cuticular ornamentation consists of punctuations, which may be evenly distributed and of equal size (homogenous cuticle), or unevenly distributed, for example, enlarged in the lateral body regions or varying along the body (heterogenous cuticle). The ornamentation may also consist of rods arranged in a "basket weave" pattern.

Anterior sensilla arranged in two or three circles. Amphidial fovea a simple transverse slit, often inconspicuous, or ventrally wound spiral, located between the cephalic setae or posterior to them. Buccal cavity with dorsal tooth usually larger than ventrosublateral ones; teeth hollow or solid; denticles may be present; three nearly equal solid teeth also occur in some genera. Male monorchic with anterior testis (a synapomorphy); pre-cloacal supplements cup-shaped (never tubular), may be absent. Females with two antidromously reflexed ovaries, the anterior gonad to the right of the intestine, the posterior gonad to the left of the intestine (a synapomorphy).

**Remarks.** The family was revised by *Venekey et al. (2019)*, who provided lists of valid species for all Chromadoridae genera.

Subfamily Chromadorinae Filipjev, 1917

**Subfamily diagnosis (modified from *Tchesunov (2014)* and *Venekey et al. (2019)*)** Cuticle homo- or heterogenous, with or without lateral differentiation of larger dots. Anterior sensilla in three separate circles (6+6+4). Amphidial fovea oval, loop-shaped or transverse slit-like, sometimes difficult to be observed under light microscope. Buccal cavity usually with three subequal solid teeth (except in *Prochromadora* Filipjev, 1922 which possesses one single dorsal tooth and *Trichromadora* Kreis, 1929 with three hollow

teeth). Pharyngeal tissue not enlarged around buccal cavity. Posterior pharyngeal bulb well defined (except in *Prochromadorella* and *Trichromadora* with poorly developed bulb). Precloacal cup-shaped supplements usually present in males.

Genus *Atrochromadora* *Wieser, 1959*
= *Chromadoropsis* *Wieser, 1954* nec Filipjev, 1918

**Genus diagnosis (modified from *Tchesunov (2014)*).** Cuticle with a homogeneous punctation pattern along the entire body, with lateral differentiation of larger dots, typically arranged in longitudinal rows. Amphidial fovea clearly visible; may be cryptocircular, unispiral, multispiral or open loop-shaped, with circular or transversely oval outline. Buccal cavity usually with three solid teeth, dorsal tooth being larger than or equal to the ventrosublateral teeth. Males usually with cup-shaped precloacal supplements or without supplements.

Type species: *Atrochromadora parva* (*De Man, 1893*) *Wieser, 1954*

**Remarks**. This genus is exclusively marine. The genus diagnosis is modified here to reflect the variety of amphidial fovea shapes found in the five previously described valid species as well as presence of body cuticle lateral differentiation without longitudinal rows in *A. dissoluta* (*Wieser, 1954*) *Wieser, 1959*. In addition, the dorsal and ventrosublateral teeth may be of equal size, as in *A. tereroa* sp. nov. The type species of the genus, *A. parva* (*De Man, 1893*) *Wieser, 1954* is the only species in which the amphidial fovea was not observed in the original description. However, subsequent descriptions by *Schuurmans Stekhoven Jr & Adam (1931)* and *Wieser (1954)* do report the presence of a visible amphidial fovea. Therefore, it is assumed that this key morphological feature is present in all species of the genus.

## List of valid species

*A. denticulata* Wieser & Hopper, 1967
*A. dissoluta* (*Wieser, 1954*) *Wieser, 1959*
    = *Chromadoropsis dissoluta* *Wieser, 1954*
*A. microlaima* (de Man, 1889) *Wieser, 1959*
    = *Chromadora microlaima* de Man, 1889
    = *Chromadorella microlaima* (de Man, 1889) Wieser, 1951
    = *Chromadorina microlaima* (de Man, 1889) de Man, 1922
    = *Chromadorina parva sensu* *Schuurmans Stekhoven Jr & Adam, 1931*
*A. obscura* *Wieser, 1959*
*A. parva* (de Man, 1893) *Wieser, 1959*
    = *Spiliphera parva* de Man, 1893
    = *Chromadorina parva* (de Man, 1893) Micoletzky, 1924
    = *Chromadoropsis parva* (de Man, 1893) *Wieser, 1954*
    = *Spiliphera antarctica* Cobb, 1914

*Atrochromadora tereroa* sp. nov.
(Table 2, Figs. 3–5)
urn:lsid:zoobank.org:act:56CAF9C3-E715-4BA4-8CA2-9F6CF462F37E

**Type locality.** Kermadec region (29.6782°S, 175.0127°W), collected during RV *Tangaroa* voyage TAN2209, from the surface of DART Buoy F, originally deployed in August 2021. Specimens of *Atrochromadora tereroa* sp. nov. were recovered from filamentous green algae.

**Type material.** Holotype male (NIWA 181,659); two paratype males and four paratype females (NIWA 181660), collected on 10 August 2022.

**Measurements:** See Table 2 for detailed measurements.

**Description:** Males. Body colourless, cylindrical, tapering slightly towards both extremities. Pigment spots (ocelli) not observed. Cuticle with transverse striations and punctations; lateral differentiation consisting of 4–6 longitudinal rows of larger punctations, extending from posterior to buccal cavity to near tail tip. Eight longitudinal rows of short somatic setae, 2–3 µm long, present from posterior to secretory-excretory pore to near tail tip. Cephalic region slightly rounded; lip region not distinctly set off. Inner labial papillae not observed; six short outer labial papillae on lip region, anterior to four cephalic setae, each 0.5–0.6 cbd long. Four sublateral rows of 2–3 cervical setae, each 2–5 µm long. Amphidial fovea cryptospiral, with flattened oval outline, located at level of cephalic setae. Buccal cavity funnel-shaped, with cuticularized walls, 14–15 µm deep and up to six µm wide; one dorsal and two ventrosublateral teeth, solid, strongly cuticularized, equal in size and shape, 4–5 µm long. Pharynx cylindrical, muscular, with oval- to pyriform-shaped posterior bulb; pharyngeal lumen not cuticularised. Nerve ring located at 52–62% of pharynx length from anterior. Secretory-excretory system present; pore located approximately halfway between level of nerve ring and anterior body extremity; pore and distal portion of ampulla cuticularized and surrounded by thin glandular layer; elongated renette cell located posterior to pharynx. Cardia small, short, not surrounded by intestine.

Reproductive system monorchic with single anterior outstretched testis located left relative to intestine. Sperm cells globular, 4–7 × 5–8 µm. Spicules paired, with velum, curved near proximal and distal ends, tapering distally, 1.0–1.1 cloacal body diameters long. Gubernaculum funnel-shaped, strongly dilated distally and denticulated. Ejaculatory glands not observed. Two conspicuous sup-shaped precloacal supplements present, located 25–28 µm anterior to cloaca and 26−28 µm apart. One short precloacal seta present ventrally. Tail conical. Three caudal glands and spinneret present.

Females. Similar to males, but often with slightly longer tail, measuring 4.1–5.1 anal body diameters in length. Reproductive system didelphic, with two opposed and reflexed ovaries; anterior ovary to the right of intestine, posterior ovary to the left. Surface of mature eggs with numerous bumps giving distinctive rough appearance, measuring approximately 25–26 × 45–49 µm. Spermatheca not observed. Vulva situated near mid-body. Proximal portion of vagina surrounded by constrictor muscle, small vaginal glands present. Proximal portion of uterus opposite vulva not conspicuously cuticularized.

Table 2 **Morphometrics (microns) of *Atrochromadora tereroa* sp. nov.** a, body length/maximum body diameter; b, body length/pharynx length; c, body length/tail length; c', tail length/anal or cloacal body diameter; cbd, corresponding body diameter; L, total body length; V, vulva distance from anterior end of body; %V, V/total body length × 100.

| Label | Males | | | Females | | | |
|---|---|---|---|---|---|---|---|
| | Holotype | Paratypes | | Paratypes | | | |
| | M1 | M2 | M3 | F1 | F2 | F3 | F4 |
| L | 749 | 741 | 758 | 769 | 728 | 748 | 810 |
| a | 29 | 27 | 28 | 26 | 24 | 28 | 29 |
| b | 6 | 6 | 6 | 6 | 6 | 6 | 6 |
| c | 9 | 9 | 8 | 9 | 8 | 7 | 8 |
| c' | 4.1 | 3.6 | 4.0 | 4.1 | 4.7 | 5.1 | 5.1 |
| Head diam. at cephalic setae | 14 | 14 | 14 | 14 | 14 | 15 | 15 |
| Head diam. at amphids | 14 | 14 | 14 | 14 | 14 | 15 | 15 |
| Length of sub-cephalic setae | 3–4 | 2–4 | 4–5 | 4–5 | 3 | 3 | 3–4 |
| Length of cephalic setae | 7–8 | 7–8 | 7–8 | 8 | 6–7 | 6–7 | 6–7 |
| Amphid height | 2 | 2 | 2 | 2 | 3 | 2 | 2 |
| Amphid width | 4 | 5 | 4 | 4 | 4 | 5 | 4 |
| Amphid width/cbd (%) | 29 | 36 | 29 | 29 | 29 | 33 | 27 |
| Amphid from anterior end | 2 | 2 | 2 | 3 | 4 | 4 | 3 |
| SE pore from anterior | 36 | 36 | 38 | 44 | 42 | 31 | 39 |
| Nerve ring from anterior end | 81 | 70 | 68 | 83 | 84 | 65 | 75 |
| Nerve ring cbd | 23 | 23 | 24 | 25 | 25 | 24 | 24 |
| Pharynx length | 131 | 125 | 130 | 137 | 130 | 125 | 132 |
| Pharyngeal bulb diam. | 22 | 23 | 22 | 23 | 23 | 23 | 24 |
| Pharyngeal bulb length | 34 | 36 | 36 | 36 | 34 | 36 | 37 |
| Pharynx cbd | 26 | 27 | 26 | 27 | 27 | 27 | 28 |
| Max. body diam. | 26 | 27 | 27 | 30 | 30 | 27 | 28 |
| Spicule length | 22 | 21 | 26 | – | – | – | – |
| Gubernacular apophyses length | 14 | 19 | 23 | – | – | – | – |
| Cloacal/anal body diam. | 21 | 22 | 23 | 21 | 20 | 20 | 19 |
| Tail length | 87 | 80 | 91 | 86 | 94 | 101 | 97 |
| V | – | – | – | 402 | 367 | 377 | 391 |
| %V | – | – | – | 52 | 50 | 50 | 48 |
| Vulval body diam. | – | – | – | 30 | 30 | 27 | 29 |

**Diagnosis.** *Atrochromadora tereroa* sp. nov. is characterised by body length of 728–810 μm, cuticle with lateral differentiation consisting of 4–6 longitudinal rows of larger punctations; cryptospiral amphidial fovea with flattened oval outline; buccal cavity with three equal solid teeth; secretory-excretory pore and distal portion of ampulla with cuticularized outline, surrounded by thin glandular layer; spicules 21–26 μm long (1.0–1.1 cbd); two cup-shaped precloacal supplements in males; and mature eggs with a distinctly rough surface due to the presence of numerous small bumps.

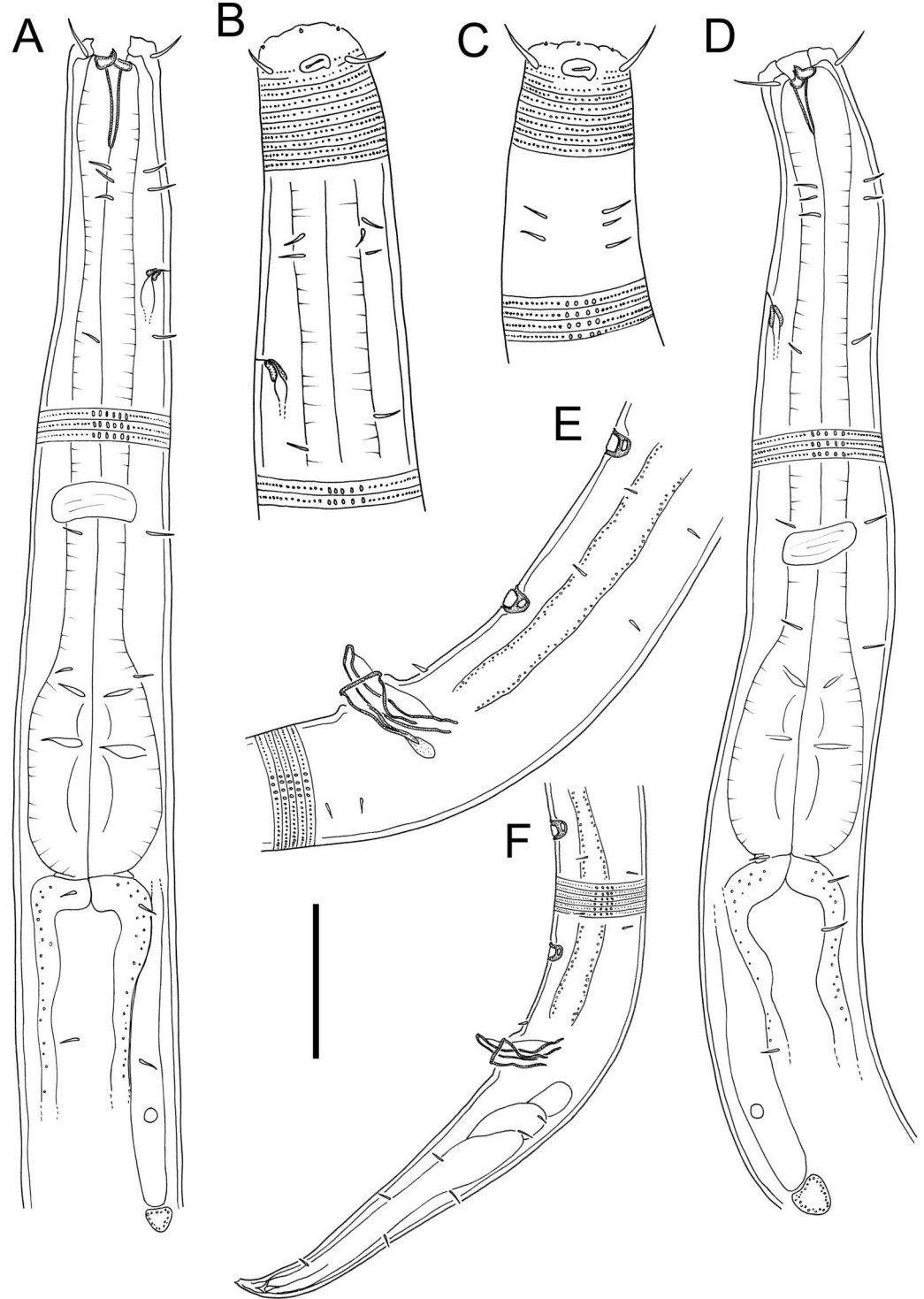

**Figure 3** *Atrochromadora tereroa* **sp. nov. drawings.** (A) Pharyngeal body region of holotype male (NIWA 181659); (B) anterior body region of female paratype (NIWA 181660); (C) anterior body region of male paratype (NIWA 181660); (D) pharyngeal body region of female paratype (NIWA 181660); (E) copulatory apparatus of male holotype (NIWA 181659); (F) posterior body region of male paratype (NIWA 181660). Scale bar: A & D = 25 microns, B & C = 20 microns, *E* = 23 microns, F = 36 microns.

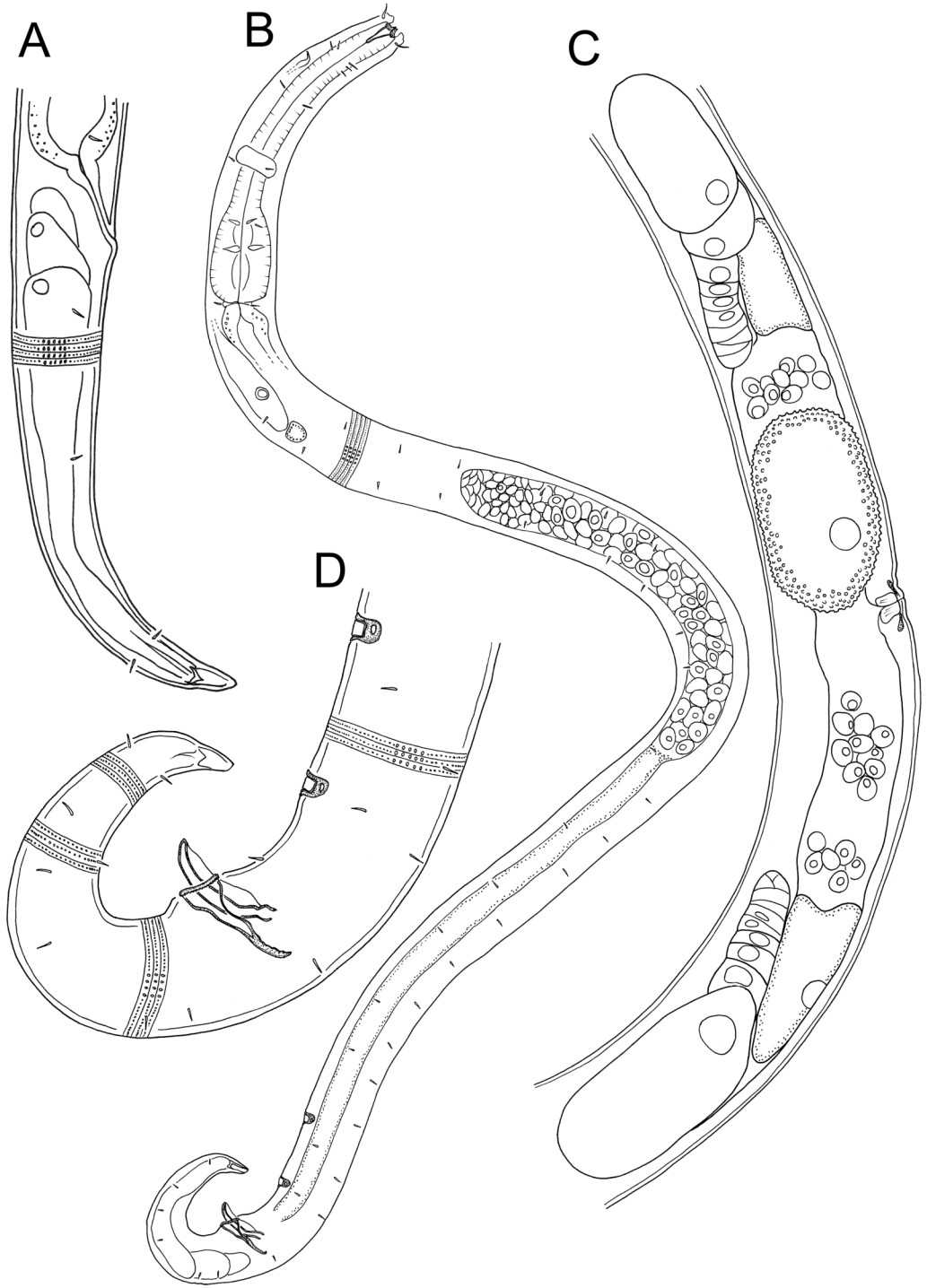

**Figure 4** *Atrochromadora tereroa* **sp. nov. drawings.** (A) Posterior body region of female paratype (NIWA 181660); (B) entire male paratype (NIWA 181660); (C) reproductive system of female paratype (NIWA 181660); (D) posterior body region of male paratype (NIWA 181660). Scale bar: A = 50 microns, B = 100 microns, C = 60 microns, D = 40 microns.

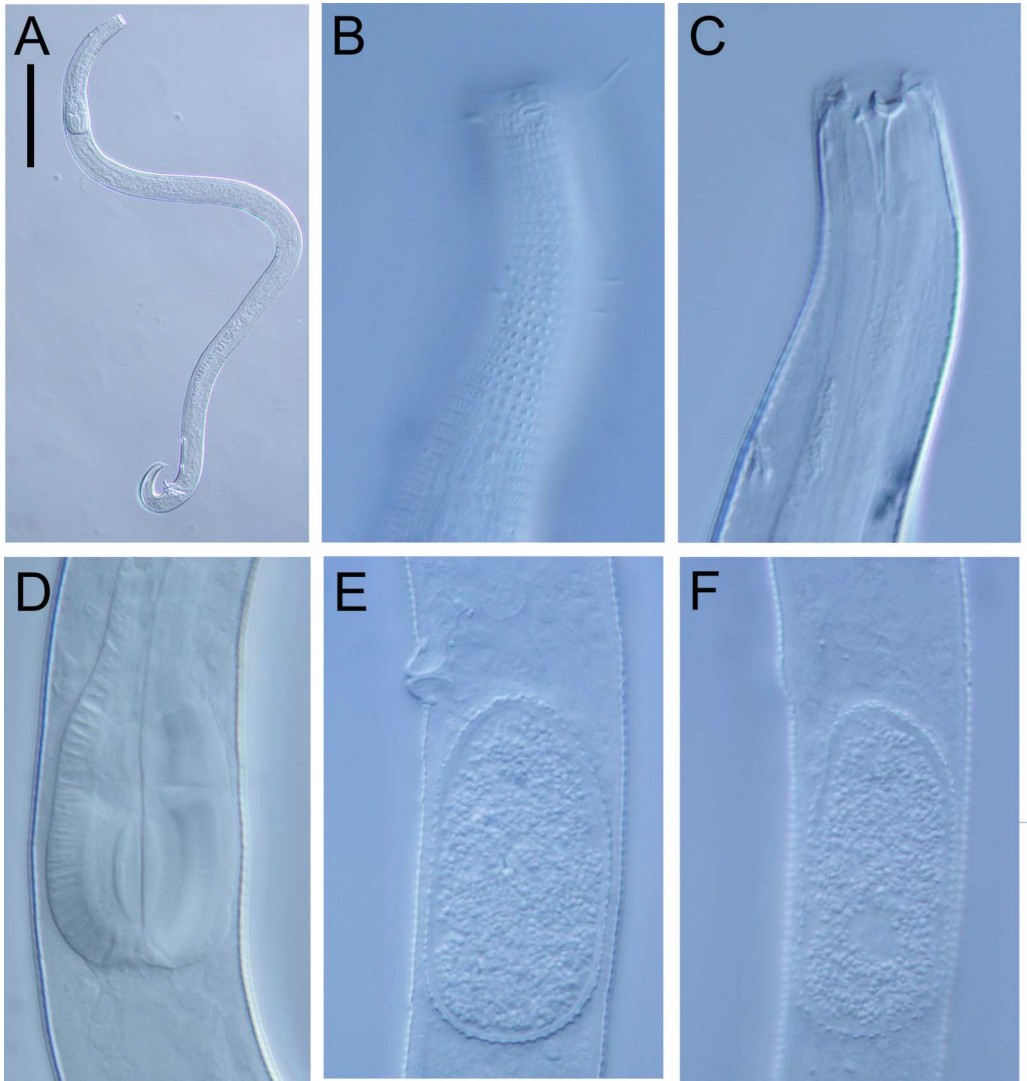

**Figure 5** *Atrochromadora tereroa* **sp. nov. light micrographs.** (A) Entire male paratype (NIWA 181660); (B) surface view of female paratype anterior body region (NIWA 181660); (C) optical cross-section of female paratype anterior body region (NIWA 181660); (D) pharyngeal bulb of male holotype (NIWA 181659); (E & D) mature egg and vulva of female paratype (NIWA 181660). Scale bar: A = 100 microns, B & C = 11 microns, D = 12 microns, E & F = 14 microns.

**Differential diagnosis.** The new species can be distinguished from all other species of the genus in having two precloacal supplements and a buccal cavity with equal-sized teeth. Other congeners possess a buccal cavity with subequal teeth and either no precloacal supplements or at least eight.

**Etymology.** The species name is a noun in apposition, derived from te reo Māori terms 'tere' (meaning to float, drift, swim, flow, glide) and 'roa' (meaning a long time) referring to the presumed ability of this species to disperse over long distances.

**Key to valid *Atrochromadora* species:**

1    Precloacal supplements absent …2
     Precloacal supplements present …3
2    Body length approximately 550 μm or less, male without ventral cuticular swelling on tail …***A. parva***
     Body length greater than 700 μm, male with ventral cuticular swelling on tail …***A. denticulata***
3    More than nine precloacal supplements present …4
     Fewer than nine precloacal supplements present …5
4    Ten precloacal supplements; spicules 26 μm long; loop-shaped amphid with oval outline …***A. obscura***
     Thirteen to fifteen precloacal supplements; spicules 35–36 μm long; multispiral amphid with circular outline …***A. microlaima***
5    Eight precloacal supplements; body length 540–770 μm; spiral amphid with round outline; buccal cavity with subequal teeth …***A. dissoluta***
     Two precloacal supplements; body length 728–810 μm; unispiral amphid with oval outline; buccal cavity with equal teeth …***A. tereroa*  sp. nov.**

Subfamily Euchromadorinae Gerlach & Riemann, 1973

**Subfamily diagnosis (from *Tchesunov (2014)* and *Venekey et al. (2019)*)**  Cuticle usually with complex, heterogenous ornamentation. The six outer labial and four cephalic setiform sensilla may be arranged in a single circle (6+10) or two distinct circles (6+6+4). Amphidial fovea transverse slit-like or oval (elliptical). Buccal cavity with large or small dorsal tooth, with or without denticles or smaller ventrosublateral teeth. Pharynx with or without defined terminal bulb. Gubernaculum usually with hammer- or L-shaped lateral pieces (erroneously referred to as telamon in some descriptions). Precloacal supplements absent in males, but a precloacal differentiation of body cuticle may be present.

**Remarks.** The subfamily was recently revised by *Datta & Al-Helal (2023)*

Genus *Euchromadora* de Man, 1886

**Genus diagnosis (modified from *Tchesunov (2014)*).**
Cuticle complex and heterogeneous, composed of hexagonal or ovoid punctuations in the anterior and posterior regions, with narrower markings confined to the lateral surface along the mid-body. Transversally elliptical amphidial fovea without surrounding cuticle fringe. Six outer labial sensilla and four cephalic sensilla setiform, arranged in separate circles. Buccal cavity with large dorsal tooth, ventrosublateral teeth, and rows of denticles. No distinct pharyngeal bulb. Gubernaculum with prominent hammer or L-shaped lateral pieces.

**Remarks.** Ten valid *Euchromadora* species are listed in the review of the family Chromadoridae by *Venekey et al. (2019)*. *Euchromadora gaulica* Inglis, 1962 may need

to be synonymized with *E. tokiokai* Wieser, 1955 due to overlap in several key body measurements particularly the strong resemblance in the structure of the copulatory apparatus, which is the main diagnostic character differentiating species of the genus.

Type species: *Euchromadora vulgaris* (Bastian, 1865) de Man, 1886

## List of valid species

*E. atypica* Blome, 1985
*E. eileenae* Inglis, 1969
*E. ezoensis* Kito, 1977
*E. gaulica* Inglis, 1962
    = *E. chitwoodi* Coles, 1965
    = *E. tridentata sensu* Wieser, 1951
*E. meadi* Wieser & Hopper, 1967
*E. permutabilis* Wieser, 1954
*E. robusta* Kulikov, Dashchenko, Koloss & Yushin, 1998
*E. striata* (Eberth, 1863) de Man, 1886
    = *E. gaulica sensu* Inglis, 1962
    = *Odontobius striatus* Eberth, 1863
*E. tokiokai* Wieser, 1955
*E. vulgaris* (Bastian, 1865) de Man, 1886
    = *Chromadora vulgaris* Bastian, 1865

*Euchromadora rebeccae* sp. nov.
Table 3, Figs. 6–9
urn:lsid:zoobank.org:act:B27A134F-39C5-441B-A02C-67925A6537A1

**Type material**: Holotype male (NIWA 182672); two paratype males and six paratype females (NIWA 182673), collected on 10 December 2021.

**Type locality:** New Zealand region, off East Cape (38.2002°S, 179.7690°W), collected during RV *Tangaroa* voyage TAN2114, from the surface of DART Buoy C, originally deployed in December 2019. Specimens of *Euchromadora rebeccae* sp. nov. were recovered from filamentous algae and goose barnacles.

**Measurements:** See Table 3 for detailed measurements.

**Description:** Males. Body with slight golden colouration, cylindrical, tapering slightly towards both extremities. Pigment spots (ocelli) not observed. Cuticle thickened, particularly in pharyngeal region and near tail tip (4–6 μm), thinner elsewhere (2–4 μm) with ornamentation and annulations visible from slightly posterior to cephalic setae to level of spinneret. Lozenge-shaped or hexagonal structures visible in cephalic and pharyngeal regions, morphing into tightly packed rectangular structures or bars sometimes with lateral differentiation of punctations in the posterior pharyngeal, mid-body and anal regions, reverting to lozenge structures in the tail region. Eight longitudinal rows of somatic setae,

**Table 3 Morphometrics (microns) of *Euchromadora rebeccae* sp. nov.** a, body length/maximum body diameter; b, body length/pharynx length; c, body length/tail length; c', tail length/anal or cloacal body diameter; cbd, corresponding body diameter; L, total body length; V, vulva distance from anterior end of body; %V, V/total body length × 100.

| | Males | | | Females | | | | | |
|---|---|---|---|---|---|---|---|---|---|
| | Holotype | Paratypes | | Paratypes | | | | | |
| Label | M1 | M2 | M3 | F1 | F2 | F3a | F3b | F4a | F4b |
| L | 1,748 | 1,532 | 1,237 | 2,136 | 1,932 | 1,764 | 1,919 | 1,797 | 2,137 |
| a | 30 | 28 | 25 | 27 | 23 | 22 | 25 | 24 | 27 |
| b | 7 | 6 | 5 | 6 | 6 | 6 | 6 | 6 | 7 |
| c | 10 | 9 | 8 | 10 | 9 | 9 | 10 | 10 | 10 |
| c' | 3.5 | 3.3 | 3.1 | 4.5 | 4.1 | 4.2 | 3.9 | 4.2 | 4.6 |
| Head diam. at cephalic setae | 28 | 29 | 28 | 33 | 33 | 33 | 32 | 34 | 32 |
| Length of cephalic setae | 8–9 | 10–11 | 9–11 | 11 | 11 | 9–10 | 12–13 | 12–13 | 10–12 |
| Excretory pore from anterior | 132 | 150 | 128 | 174 | 151 | 146 | ND | 151 | 168 |
| Nerve ring from anterior end | 110 | 122 | 113 | 149 | 130 | 127 | 144 | 135 | 137 |
| Nerve ring cbd | 45 | 45 | 44 | 46 | 48 | 47 | 52 | 49 | 48 |
| Pharynx length | 260 | 266 | 231 | 342 | 325 | 299 | 309 | 306 | 327 |
| Pharyngeal diam. at base | 31 | 32 | 31 | 42 | 40 | 38 | 41 | 37 | 37 |
| Pharynx cbd at base | 50 | 49 | 46 | 58 | 60 | 56 | 61 | 55 | 56 |
| Max. body diam. | 59 | 54 | 49 | 79 | 85 | 81 | 76 | 75 | 78 |
| Spicule length (µm; %cbd) | 97 (1.8) | 104 (2.0) | 84 (1.8) | – | – | – | – | – | – |
| Gubernaculum length | 61 | 58 | 51 | – | – | – | – | – | – |
| Telamon length | 49 | 44 | 48 | – | – | – | – | – | – |
| Cloacal/anal body diam. | 53 | 51 | 47 | 48 | 50 | 46 | 48 | 45 | 46 |
| Tail length | 183 | 169 | 147 | 216 | 206 | 193 | 188 | 189 | 212 |
| V | – | – | – | 1,098 | 939 | 870 | 981 | 917 | 1,100 |
| %V | – | – | – | 51 | 49 | 49 | 51 | 51 | 51 |
| Vulval body diam. | – | – | – | 76 | 85 | 81 | 76 | 75 | 75 |

4–5 µm long, extending along entire body length. Cephalic region slightly rounded; lip region not distinctly set off. Six inner labial papillae and six outer labial papillae in separate circles on lip region; four cephalic setae, each 0.3–0.4 cbd long. Cervical setae absent. Amphidial fovea and aperture not observed. Mouth opening surrounded by twelve cuticularized rugae. Buccal cavity funnel-shaped with cuticularized walls, approximately 30 µm deep and up to nine µm wide; one large dorsal tooth (approximately five µm long) and two smaller ventrosublateral teeth, all teeth solid and strongly cuticularised. Two rows of denticles present along the ventrosublateral sectors of the buccal cavity. Pharynx cylindrical, muscular, widening gradually posteriorly but not forming true bulb; pharyngeal lumen not cuticularised. Nerve ring located at 42–49% of pharynx length from anterior end. Secretory-excretory system present, pore located slightly posterior to nerve ring; renette cell approximately 110 × 25 µm, located immediately posterior to pharynx. Cardia medium sized (7–8 µm long), not surrounded by intestine.

Reproductive system monorchic, with single anterior outstretched testis located to the right or left of intestine. Sperm cells globular, 3–4 × 5–6 µm. Spicules paired, curved,

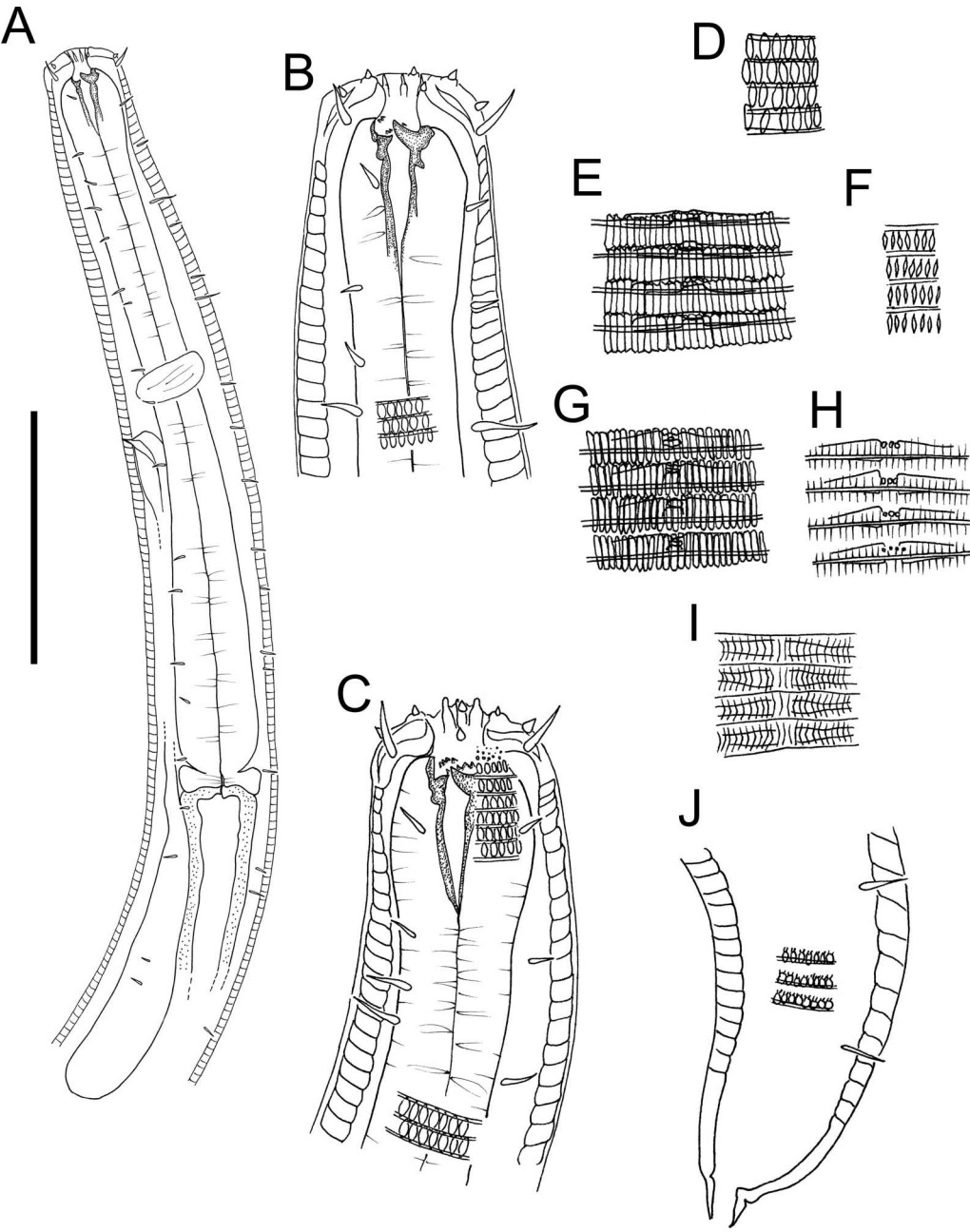

**Figure 6  *Euchromaodra rebeccae* sp. nov. drawings.** (A) Pharyngeal body region of male holotype (NIWA 182672); (B) anterior body region of male holotype (NIWA 182672); (C) anterior body region of female paratype (NIWA 182673); detail of lateral body cuticle of female paratype (NIWA 182673): (D) slightly posterior to cephalic region; (E) & (F) posterior end of pharynx (different focus); (G) & (H) mid-body (different focus); (I) anal region; (J) tip of tail. Scale bar: A = 100 microns, B & C = 50 microns, D–J = 32 microns.

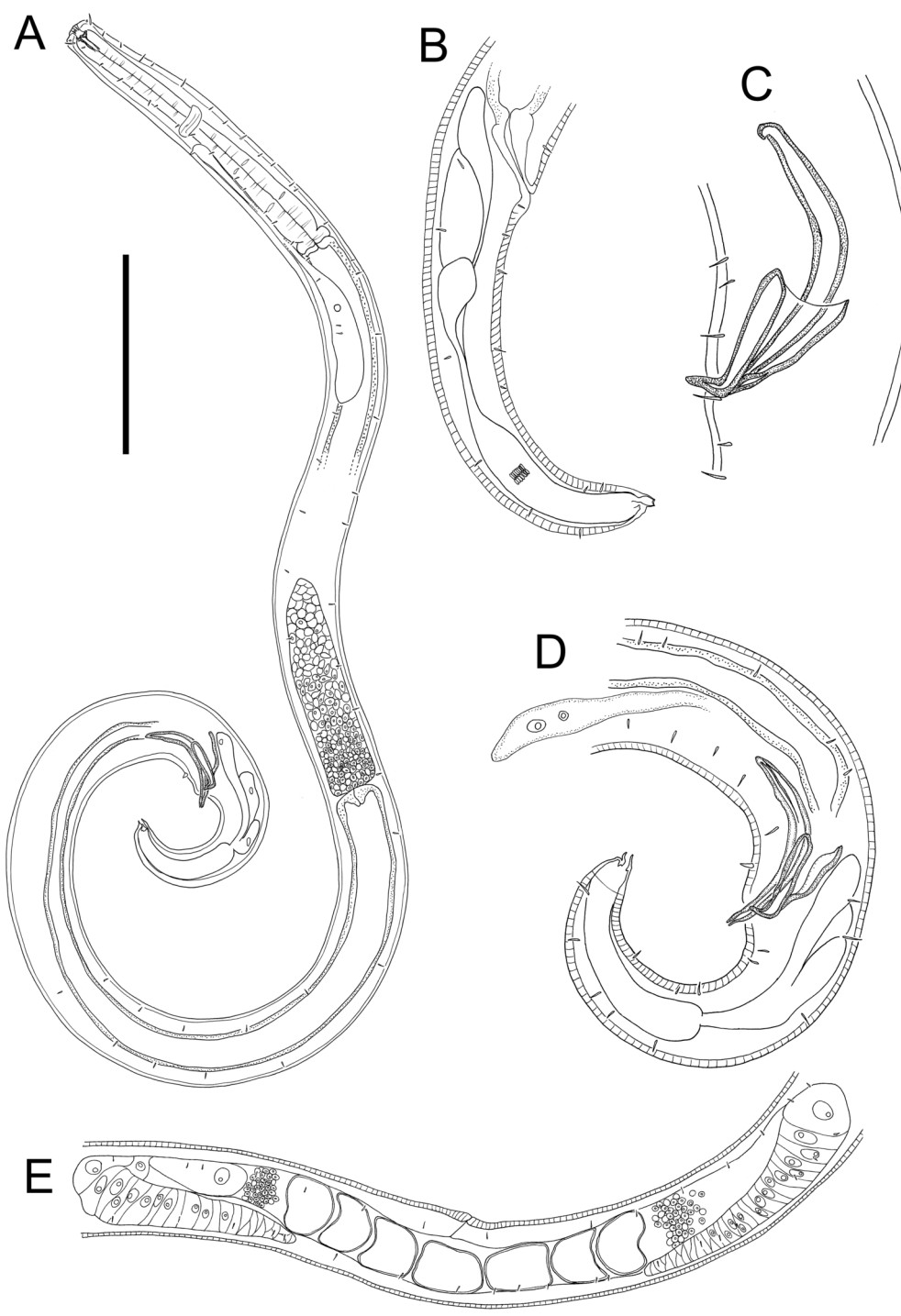

**Figure 7** ***Euchromaodra rebeccae* sp. nov. drawings.** (A) Entire male holotype (NIWA 182672); (B) posterior body region of female paratype (NIWA 182673); (C) copulatory apparatus of male paratype (NIWA 182673); (D) posterior body region of male holotype (NIWA 182672); (E) reproductive system of female paratype (NIWA 182673). Scale bar: A = 150 microns, B = 86 microns, C = 50 microns, D = 90 microns, E = 165 microns.

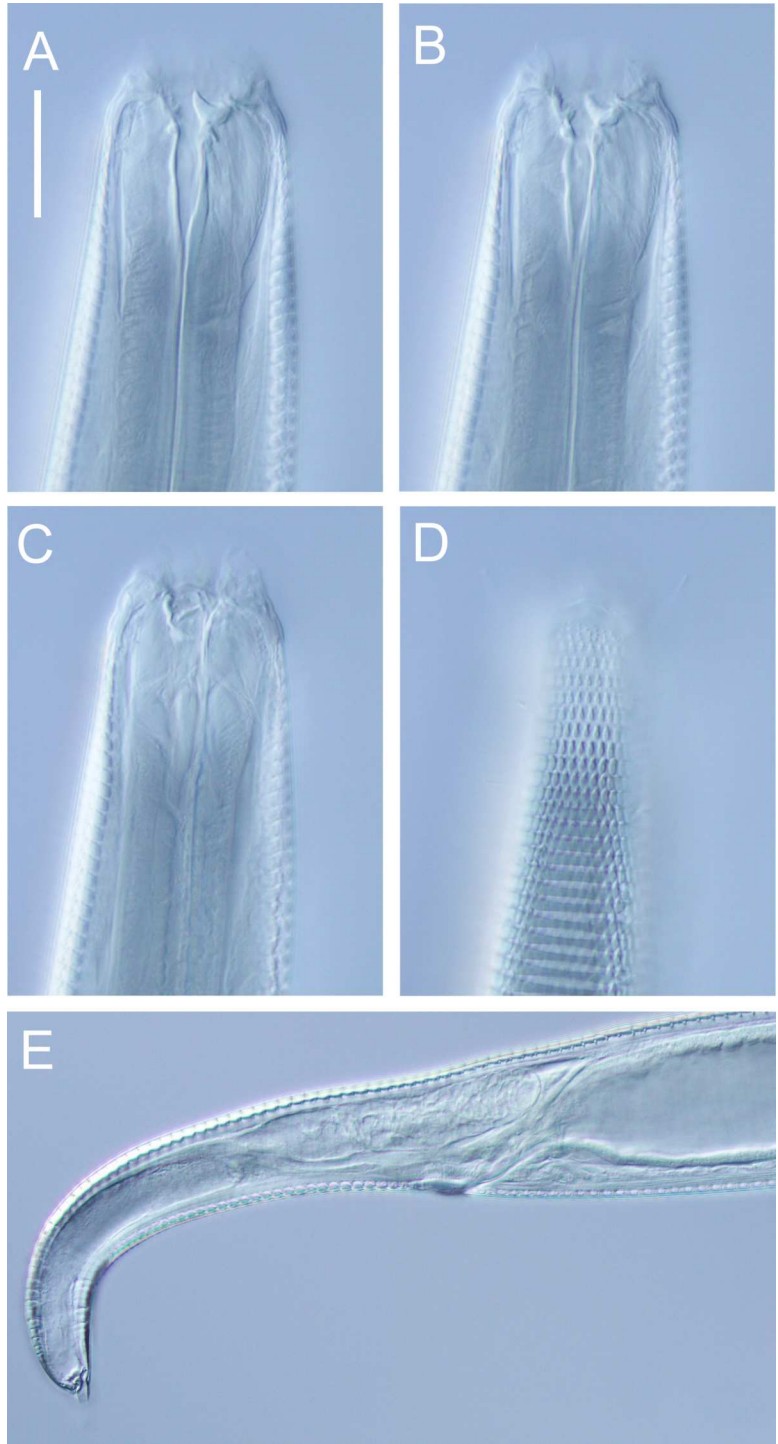

**Figure 8** *Euchromaodra rebeccae* **sp. nov. light micrographs.** (A, B, C & D) Optical cross sections and surface view of anterior body region of male paratype (NIWA 182673); (E) posterior body region of female paratype (NIWA 182673). Scale bar: A–D = 20 microns; E = 44 microns.

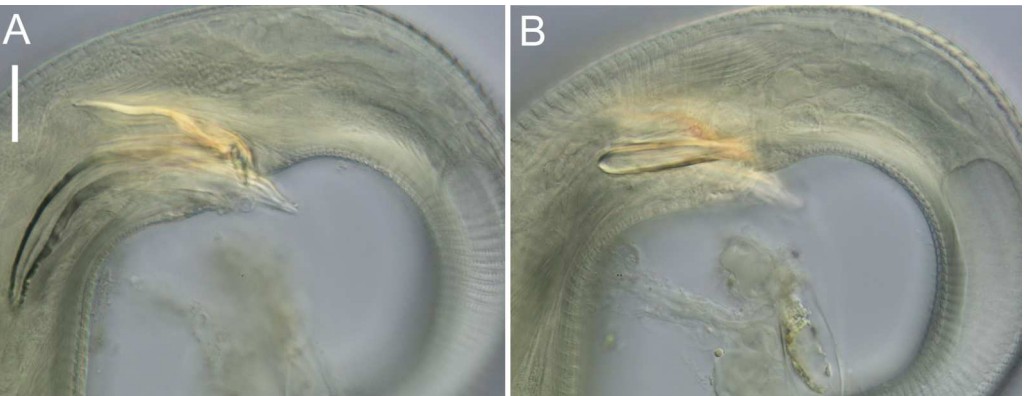

**Figure 9  *Euchromadora rebeccae* sp. nov. light micrographs.** Spicular apparatus of male holotype (NIWA 182672) showing spicules, dorsal piece of gubernaculum and distal end of telamon (A) and proximal part of telamon (B). Scale bar = 20 microns.

widest in middle portion, lacking a velum, tapering distally, measuring 1.8–2.0 cloacal body diameters in length. Gubernaculum with relatively long (51–61 μm), slightly bent dorsal piece, most strongly cuticularized along dorsal side; lateral pieces of the gubernaculum (*i.e.,* telamons) L-shaped, slightly shorter than dorsal piece (38–44 μm), without protrusions or serrations, tapered distally, rounded proximally. One pair of ejaculatory glands present, located 3–4 cloacal body diameters anterior to cloaca. Precloacal supplements absent. One short precloacal seta present ventrally, 4–5 μm long. Tail conical. Three caudal glands present; spinneret well-developed, with terminal pore.

Females. Similar to males, but often with slightly longer tail, measuring 3.9–4.6 anal body diameters in length. Reproductive system didelphic, with two opposed and reflexed ovaries; both ovaries located to the right of intestine. Mature eggs with smooth surface, measuring approximately 38–43 × 49–71 μm. Spermatheca not observed. Vulva situated near mid-body. Proximal portion of vagina without conspicuous constrictor muscle, small vaginal glands not observed. Proximal portion of uterus opposite vulva not conspicuously cuticularised.

**Diagnosis:** *Euchromadora rebeccae* sp. nov. is characterised by body length 1,237–2,137 μm, cephalic setae 0.3–0.4 cbd long, equal spicules 1.8–2.0 cloacal body diameters long, L-shaped telamons without protrusions or serrations, 38–44 μm long (0.42–0.45 of spicule length).

**Differential diagnosis:** The new species is most similar to *Euchromadora ezoensis* and *E. permutabilis* in the structure of the copulatory apparatus with equal spicules and simple L-shaped telamons without serration or protrusions. *Euchromadora rebeccae* sp. nov. differs from both species in having relatively short telamons (44–49 *vs* ≥ 54 μm in both *E. ezoensis* and *E. permutabilis.* The new species also possesses longer spicules relative to *E. ezoensis* (84–104 μm *vs* 75–85 μm), and shorter spicules relative to *E. permutabilis* (84–104 μm *vs* 104–133 μm). The new species also differs from *E. ezoensis* in having a shorter body length (1,237–2,137 *vs* 2,246–3,052 μm in *E. ezoensis*), smaller maximum body diameter (in

males: 47–59 *vs* 59–74 µm in *E. ezoensis*; in females: 75–85 *vs* 94–128 µm in *E. ezoensis*) and shorter telamon as a proportion of spicule length (0.42–0.45 *vs* 0.54–0.60 in *E. ezoensis*), and from *E. permutabilis* in having a higher ratio of a (in males: 24–27 *vs* 15–22 in *E. permutabilis*; in females: 22–24 *vs* 16–24 in *E. permutabilis*) and longer tail (in males: c' = 4.6–4.7 *vs* 2.5–3.5 in *E. permutabilis*; in females: c' = 5.0–5.2 *vs* 3.0–4.0 in *E. permutabilis*).

**Etymology.** The species is named after the author's partner, Rebecca Joy Styles.

## Dichotomous identification key of *Euchromadora* species

1 Spicules unequal in length or in shape…2
  Spicules equal in length and shape …3
2 Right spicule longer than left spicule …***E. vulgaris***
  Right spicule uniformly narrow, left spicule markedly wider but truncated and narrow proximally …***E. atypica***
3 Telamons with serrations or protrusions …4
  Telamons without serrations or protrusions …6
4 Telamons with anterior margin of distal limb serrated …***E. striata***
  Telamons with protrusions but without serration …5
5 Telamons with well-developed protrusion directed dorsocaudally at junction of distal and proximal limbs …***E. robusta***
  Telamons with distal swelling on anterior margin of distal limb …***E. eileenae***
6 Proximal and distal limbs of telamons not meeting at 90 degrees angle (telamon not L-shaped) …7
  Telamon L-shaped …8
7 Body length 1,670–2,800 µm, a = 26–40, c = 9–12…***E. gaulica***
  Body length 950–1,690 µm, a = 20–28, c ratio = 7–9…***E. tokiokai***
8 Spicules not markedly narrower than dorsal limb of telamon, dorsal portion of gubernaculum without projection …9
  Spicules uniformly slender, markedly narrower than dorsal limb of telamons, 45–56 µm long, telamon 22–24 µm long, dorsal portion of gubernaculum with proximal projection …***E. meadi***
9 Telamons total length ≥ 54 µm …10
  Telamons total length 44–49 µm, spicules length 84–104 µm, body length 1,237–2,137 µm …***E. rebeccae* sp. nov.**
10 Spicules 75–85 µm long…***E. ezoensis***
  Spicules 104–133 µm long…***E. permutabilis***

Order Monhysterida Filipjev, 1929
Family Monhysteridae de Man, 1876

**Family diagnosis (from *Fonseca & Bezerra (2014)*)** Small, slender nematodes with body lengths usually less than 2.5 mm. Body cuticle finely striated and often appearing smooth under light microscopy. Anterior sensilla in two crowns: anterior circle with six inner labial sensilla (usually papilliform), posterior circle with six outer labial sensilla

and four cephalic (usually setiform) sensilla. Amphidial fovea circular or cryptospiral and ventrally wound, varying in size (possibly due to sexual dimorphism) and in position from the anterior end. Ocelli often present in shallow-water and inland species. Buccal cavity (excluding cheilostome) surrounded by pharyngeal tissue and of varying shape: either bipartite or single V-shaped, cylindrical or minute, with or without denticles. Pharynx cylindrical, well-muscularized, sometimes slightly swollen at its anterior end and in some genera with more or less developed muscular posterior bulb. Cardia composed of a conoid part lying between pharynx and intestine, and oblong valve-like, inner part protruding into intestinal lumen. Intestine with few cells (oligocytous) arranged in two rows; dorsal and ventral. Ventral gland often present in marine and freshwater species; secretory–excretory pore from just anterior to nerve ring to the labial region. Female reproductive system monodelphic-prodelphic, with the gonad almost always outstretched on the right side of intestine. Male monorchic, spicules generally simple, of varying length, one to five times the anal body diameter. Gubernaculum of varying shape: thin without apophysis to robust with apophysis. Spermatozoa spherical. Tail conoid to elongate –conoid, similar in sexes with caudal glands opening through a single pore at the terminal spinneret; terminal setae absent.

**Remarks.** The family was revised by *Fonseca & Decraemer (2008)*, who provided lists of valid species for all Monhysteridae genera.

Genus *Halomonhystera* Andrássy, 2006

**Genus diagnosis (modified from *Tchesunov, Portnova & Van Campenhout (2015)*)**
Body stout to slender. Cuticle thin and optically smooth. Labial region not set off. Inner labial sensilla papilliform, outer labial and cephalic sensilla setiform. Amphidial fovea circular, relatively small to moderate in size, located less than one to three labial diameters from the cephalic apex. One to three lateral cervical setae present, situated at some distance posterior to the amphidial fovea; other somatic setae sparse, short and inconspicuous. Pharyngostoma cup- to funnel-shaped, small, with cuticularised walls. Pharynx cylindroid, evenly muscular throughout its length. Anteriormost stomach-like portion of the intestine (progaster) composed of four cells, set off from posterior intestine by a constriction. Ventral pore usually at labial region if discernible; ventral gland cell body large and situated at anterior intestine. Female ovary long, outstretched and located to the right of the intestine; vulva often but not always located close to the anus; posterior cuticular wall of the vagina may be thickened and cuticularised (pars refringens vaginae) closer to the vulva. Uterus of ripe females normally filled with numerous eggs and embryos; possibly most species ovoviviparous. Male gonad long, outstretched and located to the right of the intestine. Spicules slender and arcuate, slightly knobbed posteriorly. Gubernaculum with a short dorso-caudal apophysis. One midventral preanal papilla close to the cloacal opening and two or three pair of subventral papillae on the posterior half of the tail present. Three caudal glands present, two of them very conspicuous; terminal conical spinneret with an internal funnel-like structure.

Type species: *Halomonhystera disjuncta* (Bastian, 1865) Andrássy, 2006

**Remarks.** The genus was most recently revised by *Tchesunov, Portnova & Van Campenhout (2015)*. The latter authors retained *H. paradisjuncta* (De Coninck, 1943) Tchesunov, Portnova & Campenhout, 2015 as a valid species even though it had been synonymised with *H. disjuncta* by Andrássy (2006). No justification was provided for this decision, however the species is provisionally retained here pending a more thorough revision of the genus.. The diagnosis by *Tchesunov, Portnova & Van Campenhout (2015)* states that the ventral pore is located at the labial region (when discernible). However, in some species such as *H. cameroni* (Steiner, 1958) Andrássy, 2006 and *H. tangaroa* Leduc, 2014, the ventral pore is located well posterior to the buccal cavity.

*Tchesunov, Portnova & Van Campenhout (2015)* also noted that certain species of the closely-related genus *Thalassomonhystera* possess all the diagnostic characters of *Halomonhystera* except for the position of the vulva, which may be located more anteriorly relative to the anus. They stated that the position of the vulva can be in conflict with a number of other *Halomonhystera* characters. They concluded that the vulva can vary gradually in position from one species to another and does not necessarily need to be located far posteriorly (as stated in in previous diagnoses of the genus *Halomonhystera*) for a species to be ascribed to *Halomonhystera*, as long as the other characters agree with the genus diagnosis. The overlap between the genera *Halomonhystera* and *Thalassomonhystera* includes not only the position of the vulva, but also other morphological characters such as tail shape (conical in all *Halomonhystera* species and in some *Thalassomonhystera* species), buccal morphology (either simple or double in *Halomonhystera* and simple in all *Thalassomohystera* species), and amphid size (small to medium in *Halomonhystera* and small to large in *Thalassomonhystera*).

The only trait which appears to differ consistently between *Halomonhystera* and *Thalassomonhystera* as they are currently defined by *Tchesunov, Portnova & Van Campenhout (2015)* and *Fonseca & Decraemer (2008)*, respectively, is the presence of precloacal and caudal papillae in *Halomonhystera* and their absence in *Thalassomonhystera*. This difference was not discussed by *Tchesunov, Portnova & Van Campenhout (2015)*, however, given the overlap in other key morphological characteristics previously used to distinguish between the two genera (*i.e.,* the position of the vulva and buccal morphology in particular), it appears that the presence or absence of pre- and postcloacal papillae constitutes the best available character to differentiate *Halomonhystera* from *Thalassomonhystera*. According to this new definition, *Thalassomonhystera refringens* (Bresslau & Schuurmans Stekhoven, 1933) Jacobs, 1987 and *T. anoxybiotica* (Jensen, 1986) Jacobs, 1987 need to be transferred to *Halomonhystera* as they both possess precloacal and caudal papillae.

*Halomonhystera zhangi* (*Li, Huang & Huang, 2024*) was recently described from coastal *Sargassum* in the Yellow Sea, and is morphologically identical to *H. refringens* (Bresslau & Schuurmans Stekhoven, 1933) comb. nov. in most key characteristics, including body length, body ratios (a, b, c and c'), size and arrangements of anterior sensilla, position of vulva (relatively far anteriorly for the genus), amphid size and position, stoma shape, and presence and position of pre- and postcloacal papillae (*Li, Huang & Huang, 2024*). The

only slight inconsistencies are slightly longer spicules in *H. zhangi* (41–45 *vs* 39–40 µm in *H. refringens*) and opening of secretory-excretory system just posterior to level of cephalic setae (*vs* further posteriorly in *H. refringens*). The latter may be an erroneous observation; this feature can be difficult to observe and photomicrographs of the holotype specimen appear to show the secretory pore at same level as the ampulla, as indicated by a slight bulge on cuticle (Fig. 2A in *Li, Huang & Huang, 2024*). On balance, I suggest that *H. zhangi* be synonymised with *H. refringens*.

A tabular key to *Halomonhystera* species updated from *Tchesunov, Portnova & Van Campenhout (2015)* is provided in Tables S1 and S2.

### List of valid *Halomonhystera* species:

*H. anoxybiotica* (Jensen, 1986) **comb. nov.**
 = *Monhystera anoxybiotica* Jensen, 1986
 = *Thalassomonhystera anoxybiotica* (Jensen, 1986) Jacobs, 1987
*H. antarctica* (Cobb, 1914) Andrássy, 2006
 = *Monhystera antarctica* Cobb, 1914
*H. bathyislandica* (Riemann, 1995) Tchesunov, Portnova & Campenhout, 2015
 = *Thalassomonhystera bathislandica* Riemann, 1995
*H. cameroni* (Steiner, 1958) Andrássy, 2006
 = *Monhystera cameroni* Steiner, 1958
*H. chitwoodi* (Steiner, 1958) Andrássy, 2006
 = *Monhystera chitwoodi* Steiner, 1958
 = *Geomonhystera chitwoodi* (Steiner, 1958) Jacobs, 1987
*H. continentalis* Andrássy, 2006
*H. disjuncta* (Bastian, 1865) Andrássy, 2006
 = *Monhystera disjuncta* Bastian, 1865
 = *Geomonhystera disjuncta* (Bastian, 1865) Jacobs, 1987
 = *Monhystera ambigua* Bastian, 1865
 = *Monhystera vivipara* Allgén, 1929
 = *Desmolaimus viviparus* Allgén, 1929
 = *Monhystera paraambigua* Allgén, 1933
 = *Monhystera paraambiguoides* Allgén, 1932
*H. fisheri* (Zekely, Sørensen & Bright, 2006) Tchesunov, Portnova & Campenhout, 2015
 = *Thalassomonhystera fisheri* Zekely, Sørensen & Bright, 2006
*H. glaciei* (Blome and Riemann, 1999) Andrássy, 2006
 = *Geomonhystera glaciei* Blome & Riemann, 1999
*H. halophila* Andrássy, 2006
*H. hermesi* Tchesunov, Portnova & Campenhout, 2015
*H. hickeyi* Zekely Sørensen & Bright, 2006
*H. islandica* (De Coninck, 1943) Tchesunov, Portnova & Campenhout, 2015
 = *Monhystera islandica* De Coninck, 1943
 = *Eumonhystera islandica* (De Coninck, 1943) Andrássy, 1981
 = *Thalassomonhystera islandica* (De Coninck, 1943) Jacobs, 1987

*H. paradisjuncta* (De Coninck, 1943) Tchesunov, Portnova & Campenhout, 2015
    = *Monhystera paradisjuncta* (De Coninck, 1943) Andrássy, 2006
    = *Geomonhystera paradisjuncta* (De Coninck, 1943) Jacobs, 1987
*H. parasitica* Poinar, Duarte & Santos Maria, 2009
*H. refringens* (Bresslau & Schuurmans Stekhoven, 1933) **comb. nov.**
    = *Monhystera refringens* Bresslau & Schuurmans Stekhoven, 1933
    = *Thalassomonhystera refringens* (Bresslau & Schuurmans Stekhoven, 1933) Jacobs, 1987
    = *Monhystera britannica* Wieser, 1951op *Wieser, 1959*
    = *Monhystera refringens britannica* Wieser, 1951
*H. rotundicapitata* (Filipjev, 1922) Tchesunov, Portnova & Campenhout, 2015
    = *Monhystera rotundicapitata* Filijev, 1922
    = *Thalassomonhystera rotundicapitata* (Filipjev, 1922) Jacobs, 1987
*H. socialis* (Bütschli, 1874) Andrássy, 2006
    = *Monhystera socialis* Bütschli, 1874
*H. tangaroa* Leduc, 2014
*H. taurica* Tsalolikhin, 2007
*H. uniformis* (Cobb, 1914) Andrássy, 2006
    = *Monhystera uniformis* Cobb, 1914
    = *Monhystera barentsi* Steiner, 1916
*H. vandoverae* (Zekely Sørensen & Bright, 2006) Tchesunov, Portnova & Campenhout, 2015
    = *Thalassomonhystera vandoverae* Zekely Sørensen & Bright, 2006

***Halomonhystera refringens* (Bresslau & Schuurmans Stekhoven, 1933) comb. nov.**
= *Monhystera refringens* Bresslau & Schuurmans Stekhoven, 1933
= *Thalassomonhystera refringens* (Bresslau & Schuurmans Stekhoven, 1933) Jacobs, 1987
= *Monhystera britannica* Wieser, 1951op *Wieser, 1959*
= *Monhystera refringens britannica* Wieser, 1951
= *Halomonhystera zhangi* *Li, Huang & Huang, 2024*
Table 4, Figs. 10–12

**Material examined**: Three males and three females (NIWA 182674), collected on 10 December 2021.

**Sampling location.** New Zealand region, off East Cape (38.2002°S, 179.7690°W), RV *Tangaroa* voyage TAN2114, collected from surface of DART Buoy C, originally deployed in December 2019. Specimens were recovered from filamentous algae and goose barnacles.

**Distribution:** Cosmopolitan. North Sea (*Schuurmans Stekhoven Jr 1935*; *Warwick, Platt & Somerfield, 1998*), Chile (*Wieser, 1956*), Washington coast (USA; *Wieser, 1959*), Japan (*Kito, 1981*), Yellow Sea (*Li, Huang & Huang, 2024*), New Zealand (present study).

**Description:** Males. Body colourless, cylindrical, tapering slightly towards both extremities. Cuticle smooth with faint striations visible in some specimens. Sparse sublateral somatic setae, 4–5 $\mu$m long, sometimes in pairs. Cephalic region slightly rounded, not set-off. Inner labial papillae not observed; six outer labial setae and four cephalic setae of similar length and in single circle, ca. 0.3 cbd long, located on lip region usually near

**Table 4 Morphometrics (microns) of *Halamonohystera* refringens (Bresslau & Schuurmans Stekhoven, 1935) comb. nov.** a, body length/maximum body diameter; b, body length/pharynx length; c, body length/tail length; c', tail length/anal or cloacal body diameter; cbd, corresponding body diameter; L, total body length; V, vulva distance from anterior end of body; %V, V/total body length × 100.

| | Males | | | Females | | |
|---|---|---|---|---|---|---|
| Label | M1 | M2 | M3 | F1 | F2 | F3 |
| L | 536 | 568 | 544 | 614 | 610 | 603 |
| a | 24 | 27 | 26 | 22 | 23 | 24 |
| b | 6 | 5 | 5 | 5 | 5 | 6 |
| c | 6 | 6 | 6 | 6 | 6 | 6 |
| c' | 4.6 | 4.7 | 4.6 | 5.1 | 5.0 | 5.2 |
| Head diam. at cephalic setae | 10 | 10 | 10 | 11 | 11 | 12 |
| Head diam. at amphids | 13 | 13 | 13 | 14 | 14 | 15 |
| Length of sub-cephalic setae | 6 | 5 | 5 | 5 | 5 | 5–6 |
| Length of cephalic setae | 3 | 3 | 3 | 3 | 3 | 3 |
| Amphid height | 3 | 3 | 3 | 3 | 3 | 3 |
| Amphid width | 3 | 3 | 3 | 3 | 3 | 3 |
| Amphid width/cbd (%) | 25 | 25 | 26 | 22 | 21 | 19 |
| Amphid from anterior end | 6 | 7 | 6 | 6 | 6 | 8 |
| Secretory-excretory pore from anterior | 16 | 22 | 19 | 16 | 22 | 24 |
| Nerve ring from anterior end | 62 | 71 | 69 | 76 | 76 | 70 |
| Nerve ring cbd | 17 | 17 | 17 | 20 | 20 | 20 |
| Pharynx length | 96 | 109 | 105 | 114 | 118 | 108 |
| Pharyngeal diam. at base | 11 | 11 | 10 | 15 | 13 | 14 |
| Pharynx cbd at base | 18 | 18 | 17 | 22 | 21 | 21 |
| Max. body diam. | 22 | 21 | 21 | 28 | 26 | 25 |
| Spicule length | 40 | 39 | 39 | – | – | – |
| Gubernaculum length | 5 | 6 | 6 | – | – | – |
| Cloacal/anal body diam. | 20 | 21 | 20 | 20 | 20 | 18 |
| Tail length | 91 | 98 | 91 | 102 | 100 | 93 |
| V | – | – | – | 364 | 362 | 355 |
| %V | – | – | – | 59 | 59 | 59 |
| Vulval body diam. | – | – | – | 27 | 26 | 23 |

base. Ocelli not observed. Amphidial fovea circular with lightly cuticularized outline, medium-sized, situated ca. 0.5 cbd from anterior end. Buccal cavity funnel-shaped, with lightly cuticularized walls, 5–7 μm deep, up to four μm wide. Pharynx cylindrical, muscular, without posterior bulb; pharyngeal ducts sometimes visible. Pharyngeal lumen not cuticularised. Nerve ring at ca. 65% of pharynx length from anterior. Secretory-excretory system present; pore located at 16–18% of pharyngeal length from anterior, ampulla small, renette cell large, 10–17 × 30–32 μm, located posterior to pharynx. Cardia small, four μm long, partially surrounded by intestine; intestine of one specimen with multiple diatom frustules, 3 × 14–18 μm.

Reproductive system monorchic, with single anterior outstretched testis (though folds usually present), located to right of intestine. Sperm cells globular, ca. 2 × 2–3 μm.

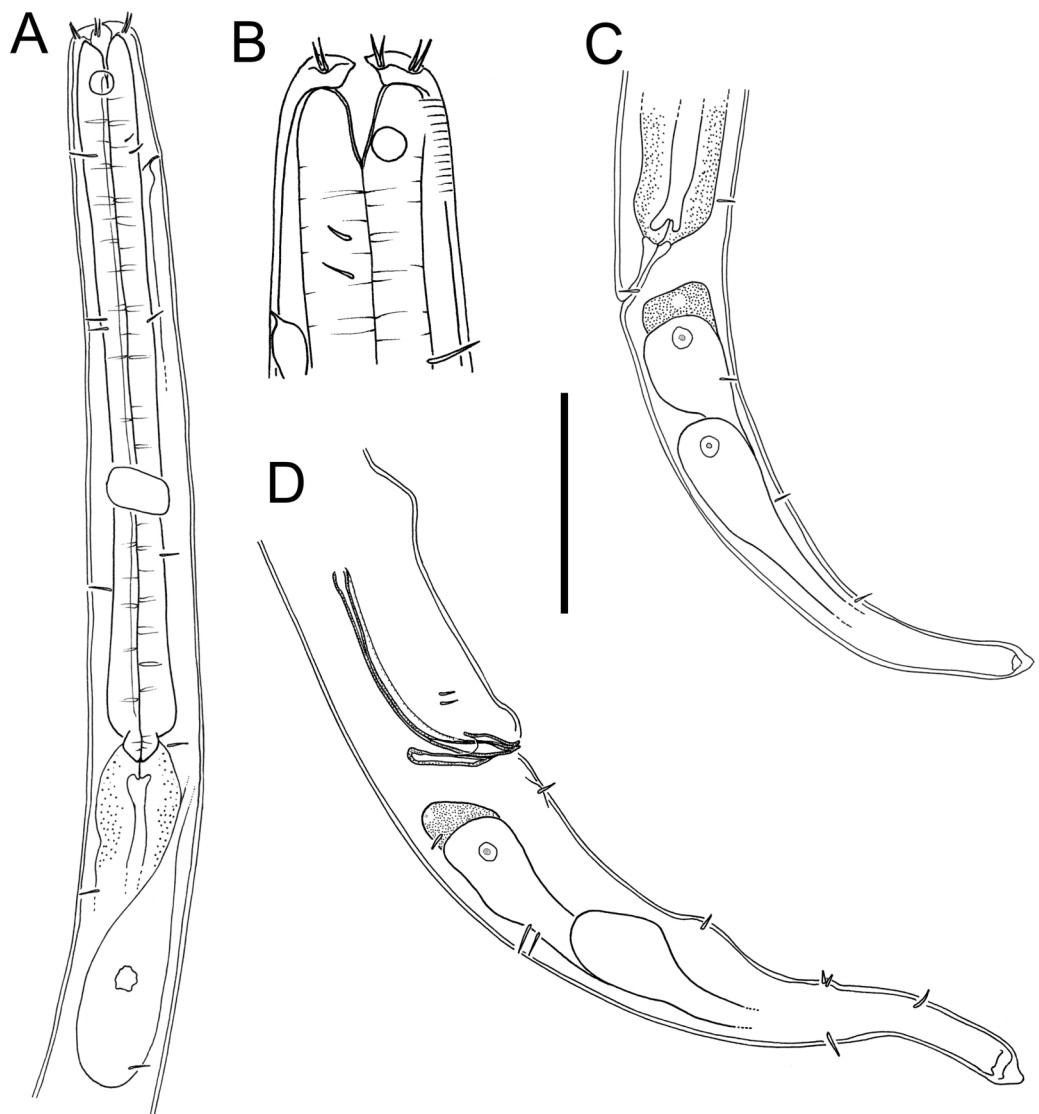

**Figure 10** *Halomonhystera refringens* **(Bresslau & Schuurmans Stekhoven, 1933) comb. nov. drawings.** (A) Pharyngeal body region of male; (B) female cephalic region; (C) female posterior body region; (D) male posterior body region. Figure 1. Scale bar: A = 35 microns, B = 20 microns, C = 40 microns, D = 30 microns.

Spicules paired, curved, with thin velum, tapering distally, 2.0 cloacal body diameters long. Gubernaculum with straight dorsal piece, without apophyses, surrounding spicules distally, ca. 15 μm long. Precloacal papilla present ventrally, 32–40 μm anterior to cloaca; another ventral papilla usually present immediately anterior to cloaca. Postcloacal papillae located 7–9, 37–42 and 57–60 μm posterior to cloaca. Anteriormost postcloacal papilla consist of pair of subventral papillae, not always distinct, each bearing one short (two μm) seta; second ventral postcloacal papillae most conspicuous, bearing pair of short (two μm) setae; posteriormost ventral postcloacal papilla bearing two pairs of short (two μm) setae.

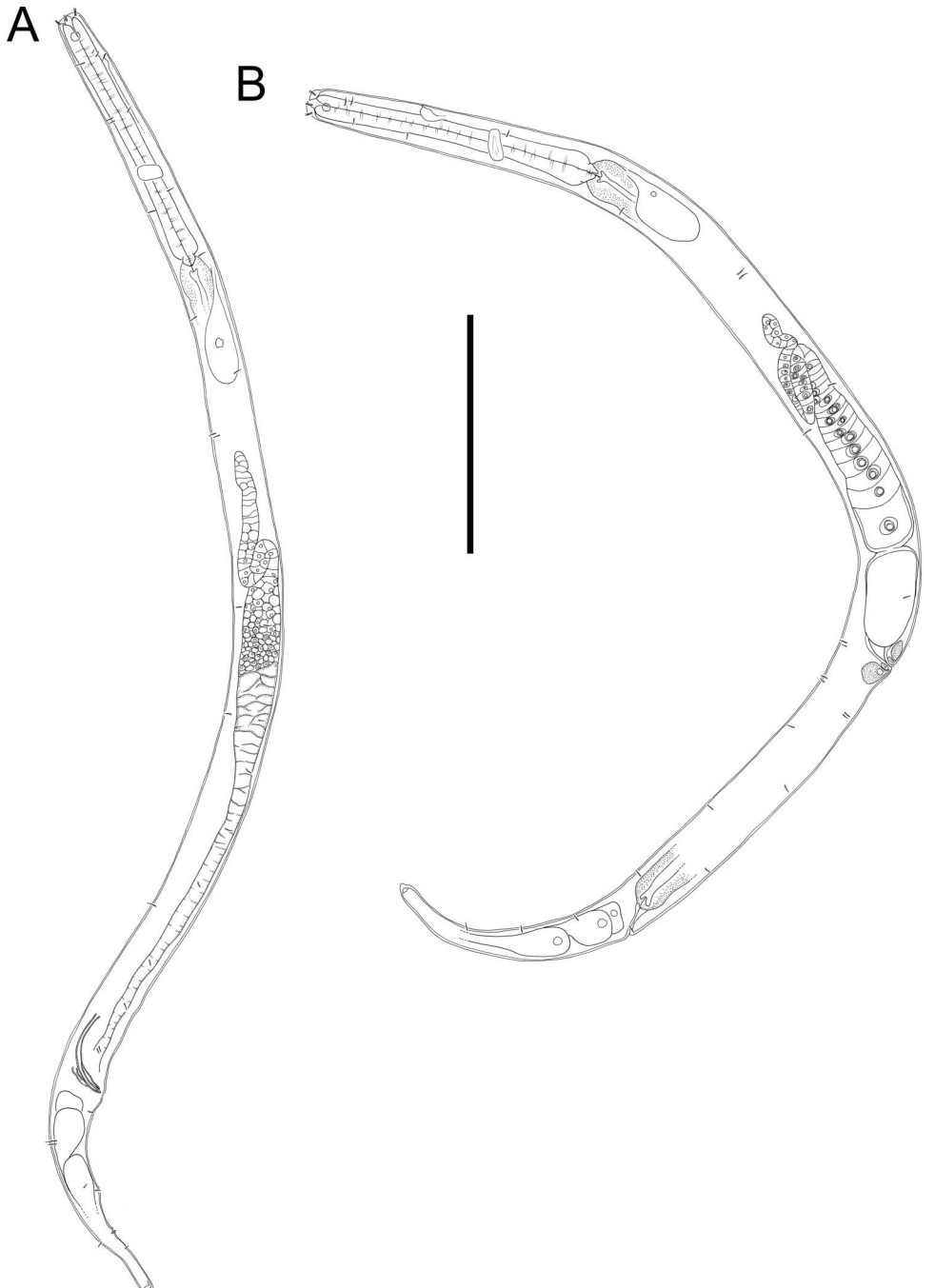

**Figure 11** *Halomonhystera refringens* **(Bresslau & Schuurmans Stekhoven, 1933) comb. nov. line drawings.** (A) Entire male; (B) entire female. Scale bar = 100 microns.

Tail conical, with short terminal cyclindrical portion and few short (3–5 μm) and sparse subdorsal setae. Three caudal glands and spinneret present.

Females. Similar to males, but with slightly smaller amphids and slightly longer tail. Reproductive system monodelphic with single anterior outstretched ovary (though fold

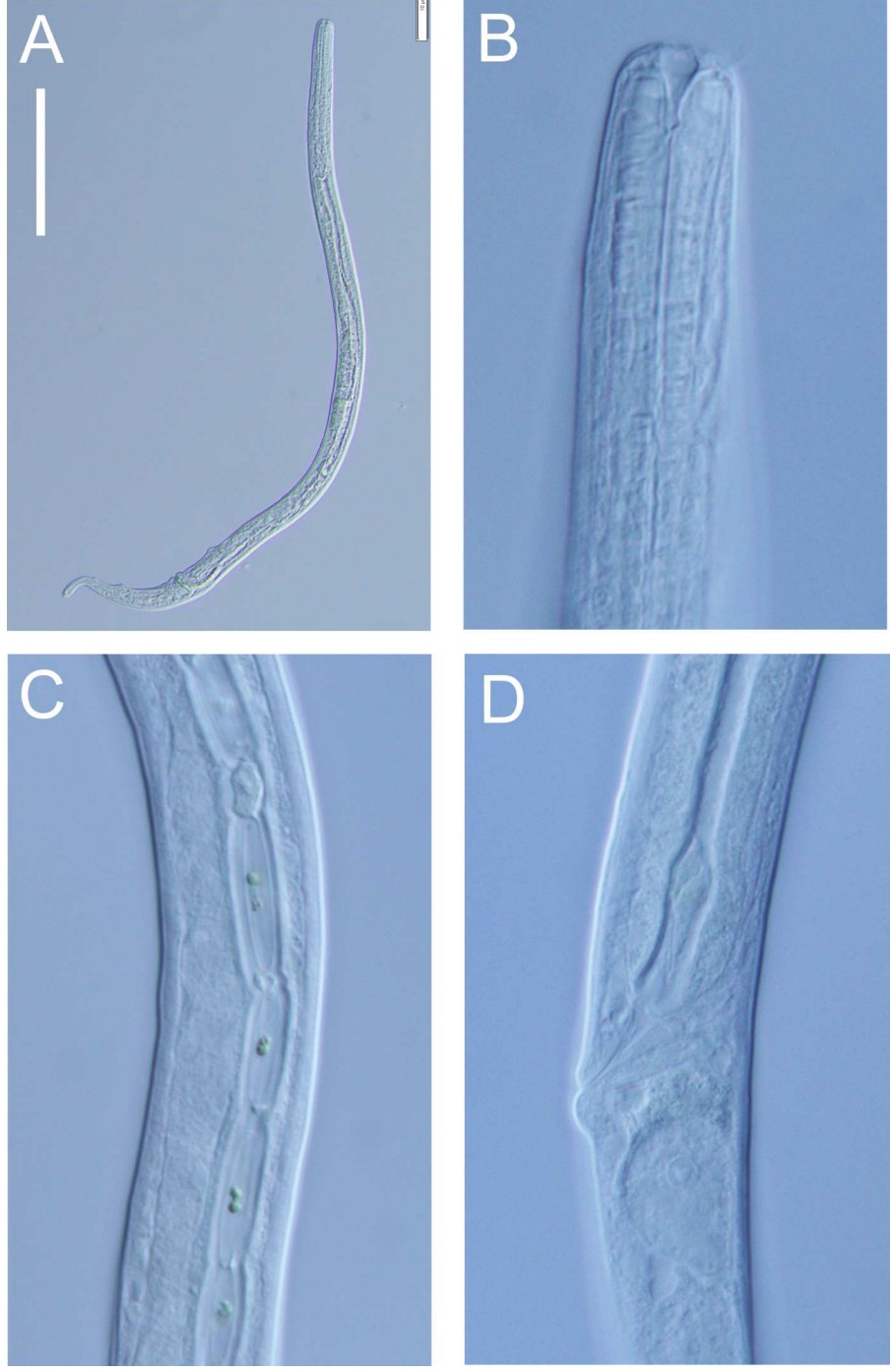

**Figure 12** *Halomonhystera refringens* **(Bresslau & Schuurmans Stekhoven, 1933) comb. nov. light micrographs.** (A) Entire male; (B) male anterior body region; (C) male intestine with several diatoms; (D) female anal body region. Scale bar: A = 100 microns, B = 13 microns, C & D = 18 microns.

usually present), located to the right of intestine; mature eggs ca. 20 × 40 μm. Spermatheca not observed. Vulva situated slightly posterior to mid-body. Proximal portion of vagina surrounded by constrictor muscle; vaginal glands present.

**Remarks.** The *Halomonhystera refringens* (Bresslau & Schuurmans Stekhoven, 1933) comb. nov. specimens from DART Buoy C off New Zealand's East Cape, agree well with the original description of the species based on material from the North Sea (*Schuurmans Stekhoven Jr 1935*). The main difference is the longer spicules in the New Zealand specimens (2.0 cloacal body diameters) compared to those from the North Sea (1.45 cloacal body diameters; *Schuurmans Stekhoven Jr 1935*, *Bresslau & Schuurmans Stekhoven Jr, 1940*) and also Chile (1.25 cloacal body diameter; *Wieser, 1956*). In contrast, descriptions based on specimens from Puget Sound (Pacific coast, USA) and Japan report spicule lengths similar to the New Zealand specimens (35–41 μm, or approximately 2.0 cloacal body diameters, as inferred from published illustrations).

Previous descriptions of *H. refringens* consistently note the presence of a precloacal papilla, along with the two posteriormost ventral postcloacal papillae. The pair of subventral postcloacal papillae, each bearing a single short seta, as described here for the New Zealand material, has not been explicitly mentioned in earlier accounts. However, these structures are not always clearly visible depending on the orientation of the specimen, and all previous descriptions do refer to the presence of setae associated with these papillae.

## DISCUSSION

The presence of nematodes on buoys deployed >100 km from the nearest landmass and in deep waters (>3,500 m water depth) shows that some nematode species are capable of dispersing over long distances to colonise new substrates. *Halomonhystera* is an opportunistic genus with the ability to colonise a wide range of habitats from intertidal seaweeds to ship hulls, food falls, cold seeps and hydrothermal vents (*Ólafsson, 1992*; *Flint et al., 2006*;, *Van Gaever et al,, 2006*; *Chan, MacIsaac & Bailey, 2016*). *Halomonhystera refringens* (Bresslau & Schuurmans Stekhoven, 1933) comb. nov. has a cosmopolitan distribution consistent with the ability for long distance dispersal. The closely related species *H. disjuncta* is also cosmopolitan but molecular studies have shown it to be a species complex comprising several distinct species (*Derycke et al., 2007*; *Fonseca, Derycke & Moens, 2008*). The presence of several diatom in the intestine of *H. refringens* (Bresslau & Schuurmans Stekhoven, 1933) comb. nov. shows that this species is able to feed on microalgae that grow among the filamentous seaweed that cover the buoys. *Halomonhystera disjuncta* has also been shown to be able to feed on diatoms and other algae in experimental settings (*Moens & Vincx, 1997*).

*Euchromadora* species are often found living on macroalgae, for example *E. ezoensis* on subtidal *Sargassum confusum* (*Kito, 1977*), *E. robusta* on shallow green and brown algae (*Kulikov et al., 1998*) and *E. eileenae* on kelp holdfasts (*Inglis, 1969*). Likewise, *Atrochromadora* species such as *A. dissoluta* (*Wieser, 1954*) and *A. parva* (*De Man, 1893*) are frequently associated with algal substrates. This habitat preference likely facilitates long distance dispersal *via* drifting macroalgal fragments (*Ptatscheck & Traunsperger, 2020*).

*Atrochromadora tereroa* sp. nov. is the second species of the genus recorded from the New Zealand region; the first species, *Atrochromadora parva*, was recorded from the coast of Campbell Island by *Allgén (1932)*. This is the first species of the genus *Euchromadora* to be recorded from the New Zealand region.

**Anatomical Abbreviations**

| | |
|---|---|
| **a** | body length divided by maximum body diameter |
| **b** | body length divided by pharynx length |
| **c** | body length divided by tail length |
| **c'** | tail length divided by anal or cloacal body diameter |
| **cbd** | corresponding body diameter |
| **L** | total body length |
| **n** | number of specimens |
| **V** | distance from anterior end to vulva |
| **%V** | V/total body length $\times 100$ |

# ACKNOWLEDGEMENTS

I thank the New Zealand National Emergency Management Agency for the opportunity to collect biological samples on service voyages for the New Zealand Tsunami Detection network, and Rachael Peart for collecting the biofouling samples at sea. I extend my gratitude to the Ngāti Kuri Trust Board, Jerry Norman and Te Tira Tapaina Ingoa for their support and guidance and for sharing their mātauranga o Ngāti Kuri.

## Funding

Funding for this work was provided by NIWA's Marine Invertebrate Taxonomy Project OCBR2402 (Protecting Marine Biodiversity programme). The funders had no role in study design, data collection and analysis, decision to publish, or preparation of the manuscript.

## Grant Disclosures

The following grant information was disclosed by the author:
NIWA's Marine Invertebrate Taxonomy Project: OCBR2402.

## Competing Interests

The authors declare there are no competing interests.

## Author Contributions

- Daniel Leduc conceived and designed the experiments, performed the experiments, analyzed the data, prepared figures and/or tables, authored or reviewed drafts of the article, and approved the final draft.

## Field Study Permissions

The following information was supplied relating to field study approvals (i.e., approving body and any reference numbers):

This work was conducted under Ministry for Primary Industries Special Permit 666-9.

## Data Availability

The raw data on nematode morphometrics are available in the tables and Supplementary File.

## New Species Registration

The following information was supplied regarding the registration of a newly described species:

Publication LSID:

urn:lsid:zoobank.org:pub:12C307BD-8D44-492C-AB65-673315A31097

Atrochromadora tereroa sp. nov.:

urn:lsid:zoobank.org:act:56CAF9C3-E715-4BA4-8CA2-9F6CF462F37E

Euchromadora rebeccae sp. nov.:

urn:lsid:zoobank.org:act:B27A134F-39C5-441B-A02C-67925A6537A1.

## Supplemental Information

Supplemental information for this article can be found online at http://dx.doi.org/10.7717/peerj.19789#supplemental-information.

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
