# Peer review of "New and known free-living nematode species (Nematoda: Chromadorea) from offshore tsunami monitoring buoys in the Southwest Pacific Ocean"

_PeerJ, doi:10.7717/peerj.19789_

## Round 0.1 · original submission · Minor Revisions

Both reviewers provided brief evaluations. This is welcome news at a time when reviewers often feel compelled to write pages of comments.

The manuscript appears extremely well prepared, and I found only minor typographical errors, as mentioned by one of the reviewers.

For Figures 5 and 8, check the word "light."

I have a slight concern regarding the use of the word "micron" throughout the text. "mMicron" is not an official SI unit, and "µm" should be used.

Reviewer 1 ·

Basic reporting

There are minor misprints in several places of the manuscript. Other than that, the manuscript meets the requirements of the journal and the regulations of the International Code of Zoological Nomenclature.

Experimental design

This is the standard taxonomic description, all requirements to which are being followed.

Validity of the findings

I see no reasons to doubt the validity of the findings.

Additional comments

Line 160 and line 185-186 mentions "A. verrucosova sp. now." which I assume is a misprint for Atrochromadora tereroa sp.nov.

Is it possible to create a dichotomous key for the genus Halomonhystera?

It would be good to have photographs of the male posterior end for Euchromadora rebeccae

Abbreviations are explained both in the text of the manuscript and in the table legends. This seem to be a bit redundant.

·

Basic reporting

I have no further comments; the manuscript, in its current form, satisfies the journal’s publication criteria.

Experimental design

No concerns. The experimental design is sound, clearly presented, and effectively implemented. The research question is well defined and investigated using robust methodology.

Validity of the findings

No further comments. The results are credible and strongly backed by the data, with conclusions that are clearly aligned with the research question and appropriately confined to the findings.

Additional comments

The manuscript is well written, clearly structured, and provides valuable taxonomic information. The figures and tables are well prepared and effectively support the main text. Overall, I have no additional comments regarding the structure of the manuscript at this time. However, I have noted specific suggestions and revisions directly within the manuscript during the review process. I hope these suggestions will be carefully considered and appropriately reflected throughout the revised version.

---

## Round 0.2 · accepted · Accept

This is a nice paper on systematics of marine nematodes. A few typos are still present: see the list below. Please edit at the proof stage.
Line 253 precloaccal
Line 284 Chrmodoridae
Line 364 smallermaximum
Line 382 dosrocaudally
Line 594 cylindrical